# Tooth for a tooth: Does fighting serve as a deterrent to greater violence in the modern NHL

**Michael R. Betz** [ID]*

Department of Human Sciences, The Ohio State University, Columbus, Ohio, United States of America

* betz.40@osu.edu

## Abstract

Fighting has been part of the fabric of the NHL for nearly a century. Recent sharp declines in the frequency of fighting and increased understanding of the long-term consequences of traumatic brain injuries have led many to question whether fighting still has a place in the modern NHL. League commissioner Gary Bettman as recently as 2019 testified before Canadian Parliament that fighting has a deterrent effect, reducing the overall level of violent and dangerous plays within the game. This study empirically examines this claim and tests whether fighting indeed serves as a deterrent to undesirable behaviors in the NHL. I examine data on all regular season penalties from 2010–2019 to determine whether fighting and the threat of fighting is empirically related the level of violence in NHL games. Using a mix of descriptive and quasi-experimental approaches, I find no quantifiable evidence that fighting serves as a deterrent to undesirable violent behaviors in the NHL. To the contrary, I find that teams and players who fight are responsible for a disproportionate amount of the violent penalties that happen across the league. These results have implications for player safety in the many professional—and especially junior—hockey leagues around the world that sanction in-game fighting.

## 1. Introduction

The National Hockey League (NHL) is unique among major North American sports leagues in that it does not eject or suspend players that engage in fighting. The league has long taken the stance that fighting serves as a deterrent for more egregious behavior. Although a better understanding of the far-reaching implications of traumatic brain injuries has more recently brought fighting under greater scrutiny, the league has been unwavering in its support for fighting's role in the game. While testifying before the Canadian Parliament about concussions in sport in 2019, NHL commissioner Gary Bettman was pressed about the feasibility of players policing themselves through fighting. Bettman responded strongly against the idea that the NHL should consider harsher penalties for fighting, citing fighting's purported ability to deter even more flagrant violence. When questioned whether empirical evidence exists to support this "deterrence" argument, Mr. Bettman replied "I don't know how you would study that"

**Data Availability Statement:** All relevant data are within the paper and its Supporting information files.

**Funding:** The author(s) received no specific funding for this work.

**Competing interests:** The authors have declared that no competing interests exist.

[1]. This study does precisely that by using information on every penalty assessed in the NHL between 2010–2019 to determine whether fighting is an effective deterrent to violent behaviors in the NHL.

In a broader context, this research question evaluates specialized versus community deterrence in sports. Previous studies have found that community rule enforcement can work well in groups small enough where individuals' behaviors can be adequately observed by other members of the community [2–4]. While the NHL had over 900 players play at least one game in the 2018–19 season, several conditions make it more likely players can adequately monitor each other. First, the average NHL player plays in the league for 5 years, giving them sufficient time to become familiar with most community members. Second, players often change teams through free agency or trade and have an opportunity to become more intimately familiar with players on other teams. This creates further network effects and a high level of information sharing between players. If a player does not have a first order relationship with any given player in the league, it is a near certainty that someone on their current team does. Third, every game is recorded and the highlights of each game are televised. NHL locker rooms have a strong culture of watching highlights and games from around the league. When another player acts in a way that violates the community norms, most of the league's players are aware of it by the next day.

Even though the NHL player community is likely small enough for community enforcement of rules and norms, several other barriers may compromise NHL players' ability to effectively enforce norms. First, players are highly biased toward their own team's interest above the interest of the NHL community as a whole. While players have a vested interest in overall league success, ultimately, they are paid by and subservient to their current team. Previous research on fighting in the NHL found that players use fighting tactically, avoiding fights in higher stakes situations [5], suggesting team interests come before those of the greater community. Additionally, skeptics of the community deterrent approach would point to the fact that fights are just as likely to occur after a hard, legal hit as they are after an illegal hit, suggesting opposing players have a biased perspective of what should be punishable by fight. It is reasonable to consider whether to entrust the players—who are so emotionally invested in the game itself—with the responsibility to administer justice in an unbiased way. Second, community deterrent skeptics point out that fighting usually only occurs between the few players on each team that are willing to fight and so are often unrelated to holding a particular player accountable for past transgressions. Indeed, 76% of NHL players did not engage in a fight during the 2018–19 season, suggesting fighting only occurs between a limited subset of players. This has become increasingly problematic for the community deterrent advocates, as several teams no longer employ such players and would raise questions about the integrity of the games in which one team does not have a player capable of filling the "enforcer" role.

Because of the unique place fighting occupies in hockey, the subject has attracted interest from researchers, and particularly so in the past two decades. Studies have examined a range of questions relevant to coaches and general managers, such as whether fighting affords teams a tactical advantage and is related to better game outcomes [6]. Others have explored league and ownership incentives to maintain support for fighting in the face of increasing understanding about the impact of head injuries [7–9], incentives for individual players to fight more [6], the conditions under which fights are more likely to occur [7, 10], and ethics of fighting [11]. DeAngelo, Humphries, and Reimers [12] use data from an earlier era in the NHL to examine the effect of fights and officiating on body checking. They found that body checking occurred at a lower right immediately after fights, but found little deterrence effect from fighting in the presence of stronger officiating. This study adds to this literature, first, by using more recent data that reflect the dramatic shift in the frequency and role of fighting in the NHL (see Fig 1).

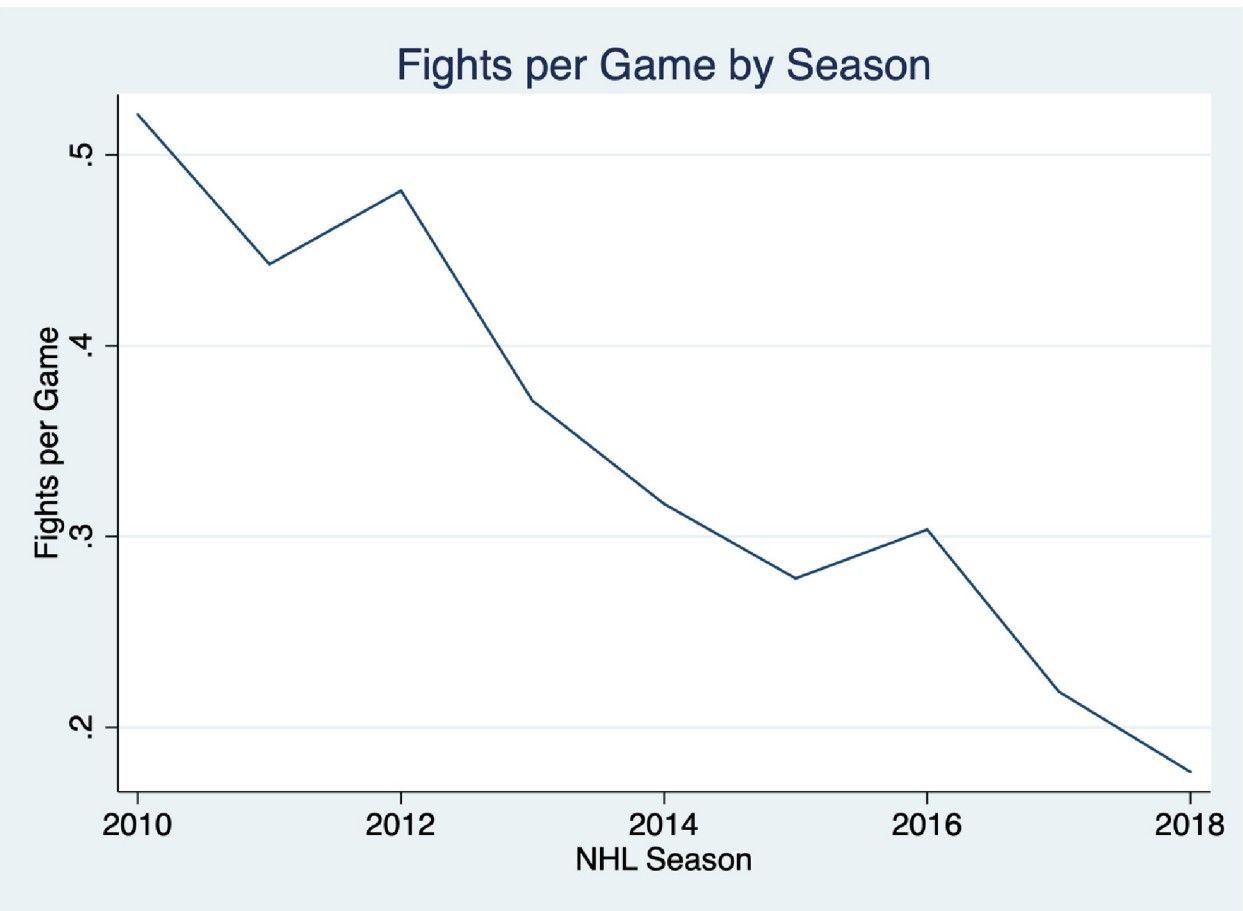

**Fig 1. NHL fights per game by season 2010–2019.** This figure shows the average number of regular season fights per game for each season between the 2010–11 and 2018–19 seasons.

Over the past decade, the NHL has largely recovered from the talent deficit created by rapid expansion in the 1990s that necessitated an influx of lesser talented—and often more violent—players into the league. The result has been that there is now a sufficient supply of skilled players where team executive can fill out their rosters with faster, more skilled players, over those that thrive on pugilism and intimidation [13] and this has had a marked effect on how the game is played. Second, by distinguishing between violent and tactical penalties as a measure of violent actions I distinguish between legal forms of body checking and those that are more violent. Third, I testing for a potential deterrence effect that persists across games by testing whether a fight in the previous game between the same teams results in less violent play the next time the two teams meet.

The results of this study have practical implications for the sport of hockey in a broader context. While fighting is heavily penalized in minor hockey, the NCAA, and international competition, many professional and junior leagues in North America and Europe follow the NHL's relatively lax stance on fight by penalizing it with a 5-minute penalty. The results are particularly important for junior hockey leagues in the U.S. and Canada, which serve as the main training grounds for players age 16–19 who aspire to the NCAA and professional ranks. Because of the greater impact of traumatic brain injuries on younger athletes, reducing such

injuries is particularly salient to those who govern these leagues. If fighting does indeed prevent greater violence, recent sanctions making fighting less common in these leagues should be rolled back. If fighting does not serve as a deterrent against more violent behaviors, then league governors must ask whether the other potential benefits of allowing fighting (i.e. fan appeal or tactical advantages) outweigh the costs.

The rest of the paper proceeds as follows: In the next section, I explain the data. The following section discusses the several empirical approaches and discusses the result. The final section offers some conclusions.

## 2. Data and methods

### 2.1 Data

For the empirical analysis, I use game-level data on all penalties in the NHL procured from ICYdata, a website devoted NHL data aggregation and analysis. These data include detailed penalty information from every NHL game including information on penalty type, offending player, and time of penalty. I use these data to distinguish between strategic and violent minor, major, and misconduct penalties as a proxy for other non-fighting violent behaviors. For minor, major, and misconduct penalties, players spend two, five, and ten minutes in the penalty box, respectively. Violent penalties are distinct from strategic penalties, which are infractions where the offending player's aim is to gain a strategic advantage over their opponent. These include penalties like tripping, hooking, holding, and minor interference penalties. I separate out violent penalties, which include infractions like boarding, charging, elbowing, roughing, and major interference penalties, which are the result of behaviors aimed at injuring or intimidating an opponent. The full set of infractions characterized as violent are included in Table 1 in S1 Appendix.

The purpose of creating counts of violent penalties per season, per team by season, and per game, is to create some measure of violent actions against the opposition. This measure does not capture all of an opponent's undesirable actions, but serves as a good proxy for the extent to which violent actions are occurring in the NHL and the extent to which opposing teams feel the liberty to physically abuse a particular team. Using each level of aggregation, I test for relationships between 1) league-wide aggregated fights and violent penalties by season; 2) aggregate team fights and violent penalties against by season; 3) changes in team fights and changes in violent penalties against from the first to the second half of the season; 4) differences in violent penalties before and after a fight within games; 5) the number of violent penalties in a game and whether there was a fight in the previous game between those teams; and 6) violent penalties against and the presence of a well-known fighter in a team's lineup. I explain the approach to testing for each of these relationships in greater detail below.

### 2.2 Methods

I begin by examining graphical depictions of annual aggregate league-wide trends in fighting and violent penalties to determine if even a simple association exists between the two trends over the study period. From there, I use more sophisticated analysis to account for potential confounding factors influencing the relationship at the team and individual levels.

To estimate whether there is a relationship between the number of fights and violent penalties at the team level, I use fixed-effect Poisson models taking the following form.

$$Penalty_{ij} = \beta_1 Fights_{ij} + \delta_i + \gamma_j + \varepsilon_{ij} \tag{1}$$

where $Penalty_{ij}$ represents the season total counts of either violent minor penalties, violent

major penalties, or violent misconduct penalties committed against team $i$ in season $j$. $Fights_{ij}$ represents the season total counts of fights for each team, $\delta_i$ and $\gamma_j$ are team- and season-fixed effects, and $\varepsilon_{ij}$ is a normally distributed error term. $\beta_1$ is the regression coefficient of interest. Robust standard errors are estimated for all models.

The next set of models remain at the team level and estimate whether teams that experienced more violent penalties against them in the first half of the season responded with more fights in the second half of the season in an effort to deter violence against them. Formally,

$$\Delta Fights_{ij,h2-h1} = \beta_1 Penalty_{ij,h1} + \delta_i + \gamma_j + \varepsilon_{ij} \tag{2}$$

where $\Delta Fights_{ij,h2-h1}$ is the change in the total number of aggregate team $i$ fights in the first half of season $j$ to the second half of season $j$, $Penalty_{ij,h1}$ is total number of either violent minor, violent major, or violent misconduct penalties, taken against team $i$ in season $j$ in the first half of season $j$.

Next, I estimate models to determine whether the negative relationship in in the models from Eq (2) stem from more a aggressive/violent overall style of play among teams that fight more often. Eq (3) below details these models.

$$Fights_{ij} = \beta_1 ViolentTaken_{ij} + \beta_2 ViolentAgainst_{ij} + \delta_i + \gamma_j + \varepsilon_{ij} \tag{3}$$

where $Fights_{ij}$ is the total number of fights for team $i$ in season $j$, $ViolentTaken_{ij}$ is the number of violent penalties of each type (either minor, major, or misconduct in their own model) team $i$ has taken in season $j$, and $ViolentAgainst_{ij}$ is the total number of violent penalties of each type. Again, $\delta_i$ and $\gamma_j$ are team and season fixed effects and $\varepsilon_{ij}$ is the error term.

I next examine models of within-game effects that compares whether violence occurs at a greater rate before or after a fight within games. To do this, I estimate a t-test for a stastically significant difference in the average violent penalties per minute before and after fights.

Finally, I exploit plausibly exogenous variation from player injuries to determine whether specific players with a well-known reputation for fighting presence (or absence) in the lineup effected the number of violent penalties the opposing team took against their team during the 2018–19 season. Eq (4) below details these player-specific models.

$$ViolentPenAgainst = \beta_1 Fighter + \delta_i + \varepsilon_i \tag{4}$$

where $ViolentPenAgainst$ is the number of violent penalties taken against the well-known fighter's team in each game, $Fighter$ is a binary variable indicating whether the well-known fighter was in the lineup for their team for that game, and $\delta_i$ is an opposing team fixed effect that controls for any conditions that might affect the number of violent penalties against a team, given their opponent. For this model, observations are at a team-game level.

## 3. Results

### 3.1 Aggregate league-wide evidence across seasons

First, I examine aggregate trends in fighting and violent penalties over the entire study period to get a sense of general penalty and fighting trends in recent years. Fig 1 shows the number of fights per game by season between the 2010–11 and 2018–19 seasons. What is most noticeable is the dramatic decline in the frequency in which fights occur over the period. Fights per game declined from 0.52 per game in the 2010–11 season to 0.18 per game in the 2018–19 season, a net decrease of about 65%. To put Fig 1 in a broader historical context, fights per game peaked in the 1987–88 season at 1.12 fighting majors per game, after which fighting in the NHL slowly declined through the 1990s and early 2000s [14], before the more rapid decline since 2010 seen

in Fig 1. The 2018–19 season ended with a fighting major per game rate of 0.18, 84% lower than the 1987–88 peak.

Fig 1 is helpful starting point in assessing the merits of a specific argument often cited by supporters of community deterrence in the NHL. The assumption is that in a counterfactual world where the players themselves could not hold other players accountable for in-game actions perceived to be dirty or outside of the unwritten norms of hockey culture, such undesirable actions would increase dramatically. In 2012 at the beginning of the sharper decline in fighting, then Toronto Maple Leafs General Manager Brian Burke, summarized this sentiment when asked about his decision to send the team's most experienced fighter to the minor leagues and his views on the reduced role of fighting in the NHL. Burke lamented the change saying "If you want a game where guys can cheap shot people and not face retribution, I'm not sure that's a healthy evolution. . .To me, it's a dangerous turn in our game. . .I wonder about the accountability in our game and the notion that players would stick up for themselves and for each other. . .the fear that if we don't have guys looking after each other that the rats will take this game over" [15]. Not only have fights per game gone down, but the proportion of games with fights has declined from 27% in 2010–11 to 15% in 2018–19. The vast majority of games with fights have only one fight. In 2018–19, of the 195 games that had at least one fight, 87% had only one fight, 1.6% had two fights, and only 5 total games had more than two fights.

If fighting were serving as a meaningful deterrent to more egregious behaviors, given the drastic decline in the fighting prevalence we would expect to see a concurrent rise in violent plays, as Mr. Burke predicted. In fact, the opposite is true. Fig 2. shows other types of penalties per game, split by whether they are violent or tactical over the same time period. What is noticeable is that all types of penalties have fallen over the period, but violent minors have fallen more than twice as fast as tactical minors (25% versus 12%). Furthermore, even though fewer penalties are being called, officiating has actually become *more strict* rather than less strict over the period. This began in the late 90s when the NHL introduced a second referee on the ice and was further prioritized coming out of the 2004 lockout in an attempt to increase scoring and high skill plays. In the past decade, the NHL has also placed additional emphasis in eliminating hits to the head [16], which has further tightened the game's officiating. It is nearly universally accepted that previous generations of players could get away with a higher degree of unpenalized violence than current players. All of this is to suggest that if the level of dirty or violent play increased (or merely remained constant) over the period due to a lower deterrence effect, we would expect to see an equally dramatic *increase* in violent minor and major penalties. Instead, we observe the precise opposite: a decline in violent minors, violent majors and misconducts. This casts some doubt on the deterrence argument, however, these trends are merely descriptive and could be driven by other unrelated leaguewide trends (e.g. increase in skill players or salary cap pressures), so more sophisticated analyses are necessary.

## 3.2 Team-level evidence

Even though predictions of widespread dirty play have not come to fruition as a result of the rapid decline in fighting in the NHL, it is possible that there was simply a league-wide oversupply of fighting, where the mere possibility of having to engage in a fight has kept dirty play to a minimum. In essence, dirty play remains low because even though fighting has declined, the threat of having to fight still exists. The claim is then that to be credible, this threat only needs to be carried out once every 5–6 games, as it was in 2018–19, rather than in every other game, as in the beginning of the period.

To address this possibility, I next examine how fighting impacts violent penalties committed against individual teams. If fighting were serving as a deterrent, we would expect teams

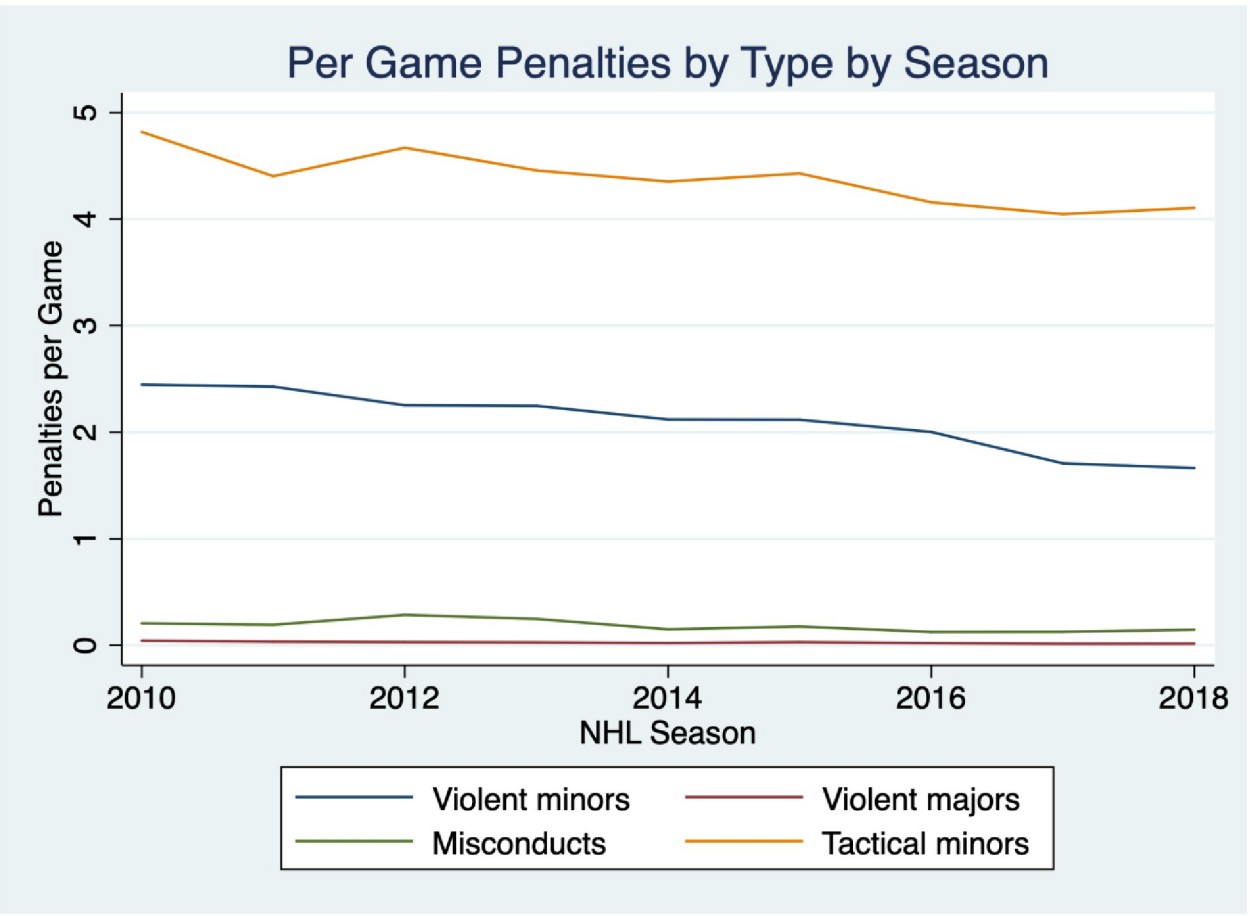

**Fig 2. Per game penalties by type by season 2010–2019.** Fig 2 shows the average number of penalties per game by season for each of tactical, violent minor, violent major, and misconduct penalties between the 2010–11 and 2018–19 seasons.

**Table 1. Marginal effect of season fights on season penalties against.**

|  | Violent Minors Against | Violent Majors Against | Violent Misconducts Against |
|---|---|---|---|
| Team's Season Fight | 0.33*** | 0.02*** | 0.15*** |
| Total | (4.12) | (10.59) | (8.28) |
| Team Fixed Effect | YES | YES | YES |
| Season Fixed Effect | YES | YES | YES |
| N | 272 | 272 | 272 |
| Pseudo R-squared | 0.11 | 0.08 | 0.15 |

Z-score in parentheses;

*p<0.05,

**p<0.01,

***p<0.001

Robust standard errors estimated

that fought more often to have fewer violent penalties committed against them. To assess this, I use season aggregate team-level data to estimate Poisson count models of violent minors, majors, and misconduct penalties against each team in each season. These models control for team-specific and season-specific fixed effects to determine whether a statistically significant relationship exists between the number of fights a team engaged in over the course of a season and the violent minors, majors, and misconduct penalties committed against that team over that season.

Table 1 shows the results of estimating Eq (1) and provides the marginal effect of an additional fight on the number of violent minors, majors, and misconducts committed against a team. In contrast to a negative relationship that would be expected if fighting is serving as a deterrent, I find that each additional fight a team engaged in was associated with 0.33, 0.02, and 0.15 *more* violent minor, major, and misconduct penalties taken against them, respectively. Instead of having fewer violent penalties taken against them, teams that fought more had more violent penalties committed against them. These models control for season-invariant and team-invariant characteristics, such as leaguewide trends in a given year or broader team playing-style philosophies that span the entire period.

The results in Table 1 suggest that teams that fight more often have more violent penalties committed against them. One possible explanation is that teams that are being physically abused by other teams feel a greater need to defend themselves, and so engage in fights more often. To test for this, I estimate linear fixed-effects models detailed in Eq (2) of the change in number of fighting majors between the first half of the season and second half of the season and include violent minors, majors, and misconducts—respectively—in the first half of the season as explanatory variables. This tests whether teams that experienced more violent penalties against them in the first half of the season responded with more fights in the second half of the season in an effort to deter violence against them.

Table 2 shows the marginal effects of these models. For each type of penalty, the coefficient is insignificant at even a more liberal 10% level, suggesting there is no statistically significant relationship between the number of violent minors, majors, and misconduct penalties committed against a team in the first half of the season and that team's change in number of fights from the first half to the second half of the season.

To further probe this relationship and test the sensitivity of the results to model specification, I estimate a parsimonious model that does not include team- or year-fixed effects in order to avoid the possibility of "over-controlling" the model. In these parsimonious models, the coefficient on the violent penalties variable remains insignificant but turns negative, suggesting that—if anything—teams that experience more violent penalties against them in the

**Table 2. Marginal effect of first half violent penalties against on difference in fighting majors from first to second half.**

|  | Violent Minors | Violent Majors | Violent Misconducts |
|---|---|---|---|
| Difference in Fighting | 0.02 | 0.26 | -0.17 |
| Major totals | (0.04) | (0.48) | (-1.00) |
| Team Fixed Effect | YES | YES | YES |
| Season Fixed Effect | YES | YES | YES |
| N | 272 | 272 | 272 |
| R-squared | 0.68 | 0.68 | 0.15 |

t-statistic in parentheses; *p<0.10, **p<0.05, ***p<0.01

Robust standard errors estimated

first half of the season fight less in the second half of the season. This result is intuitive. Violent penalties are often used to intimidate and injure more skilled opponents by less skilled opponents. The greater the skill disparity, the greater the level of violence against the more skilled team. Higher skill teams are also less likely to have players that will engage in fights, creating the negative statistical relationship between the number of fighting majors a team takes and the number of violent penalties taken against them. To summarize, teams that face more violent penalties against them do not respond by fighting more often in an effort to deter violent penalties against them and so we must explore alternative explanations to the positive relationship between fighting and violent penalties found in Table 1.

One alternative explanation is that some coaches and general managers encourage a more violent style of hockey, and so their teams engage in more fights *and* take more violent penalties against their opposition. The elevated level of violent penalties against teams that fight more may then simply be due to an in-kind response from opposing teams to the elevated level of violent play by more aggressive teams. To test this possibility, I estimate Poisson count models of team total season fights, using violent penalties against and violent penalties taken as explanatory variables for each of the penalty categories (minors, majors, misconducts, tactical).

Table 3 displays results for these models. The violent penalty taken variable is positive in all three models (Columns 1, 2, and 3) and the point estimate is statistically significant at the .01% level for violent minors and misconducts. The number of violent penalties against a team does not produce coefficient estimates statistically different from zero in Models 1 and 2. In the misconduct model, both the number of misconducts taken and misconducts against are positive statistically significant, but the marginal effect of an additional misconduct taken is 50% higher than the marginal effect of a misconduct against. Taken together, this suggests that the number of violent penalties a team takes is much more strongly related to how often one of its players fights than the amount of violent penalties the team has taken against it from its opposition. However, the number of tactical minors a team takes is unrelated to its season fight total. The lack of significance on the tactical minors coefficient suggests that teams that fight are not less disciplined and take more penalties in general, rather they are only less disciplined in the area of violent penalties. Taken together, these relationships support notion that teams that fight

**Table 3. Marginal effect of season violent penalties on season fights.**

| Dep. Variable | (1) | (2) | (3) | (4) |
|---|---|---|---|---|
| | **Violent Minors Taken** | **Violent Majors Taken** | **Misconducts Taken** | **Tactical Minors Taken** |
| Team fight total | 0.21*** | -0.07 | 0.81*** | -0.05 |
| | (5.65) | (-0.17) | (7.76) | (-1.52) |
| *Controls* | | | | |
| Violent pen. against | YES | YES | YES | YES |
| Team Fixed Effect | YES | YES | YES | YES |
| Season Fixed Effect | YES | YES | YES | YES |
| N | 272 | 272 | 272 | 272 |
| Pseudo R-squared | 0.38 | 0.36 | 0.39 | 0.36 |

Z-score in parentheses;

*p<0.05,

**p<0.01,

***p<0.001

Robust standard errors estimated

more further contribute to increased violence by initiating more violent minor and misconduct penalties.

Of course, it is possible the positive relationships in Table 3 are due to some omitted variable that could be causing a team that fights often also to have many violent minor and misconduct penalties called against them. As a placebo test, I also estimate these same models of total team fight counts, but now include tactical minors taken as a predictor variable alongside each of the three types of violent penalties, each in its own regression equation. If violent minors or misconducts are associated with some unobservable penalty-related factor that is causing the number of violent minors or misconducts taken to be spuriously related to fighting majors—for instance referee bias against a particular team because of their reputation for fighting—we would expect to see a positive and significant coefficient on the tactical minors variable in addition to the significant coefficient on violent minors and misconducts. Instead, the tactical minors coefficient is negative and insignificant (results not shown), while the coefficients on violent minors and misconducts remain positive and statistically significant. This provides additional evidence that the relationship between the amount of violent minors and misconducts a team takes and the number of times its players have fought is not simply a matter of unobserved officiating bias.

So far, I have presented evidence that there has not been a league-wide increase in violent penalties following the dramatic decline in fighting prevalence in the NHL. This relationship also holds at the team level, where teams that fight more do not prevent violent minors, majors, or misconducts against them. On the contrary, teams that fight more have *more* violent penalties against them. This result is not because teams that are physically abused are defending themselves by fighting more, if anything teams that experience more violent penalties against them fight less. Much stronger evidence exists to support the claim that teams that fight more are also the ones leveling more violence against their opponents through other violent penalties. Taken together, these results provide evidence against the hypothesis that fighting operates as a deterrent on a league-wide level or at a team level. However, its possible that aggregating penalties across the entire season may mask significant deterrent effects with particular games, a possibility I explore in the next section.

## 3.3 Within-game effects

Even though a deterrence effect is not observable when aggregating the data at the league or team level, it is possible this aggregation obscures fighting's deterrent effect within individual games. It's possible that fighting does not impact the prevalence of violent plays against a team over the course of a season, but can potentially change the behavior of the opposing team within a game. To test this hypothesis, I calculate the rate at which teams experience violent penalties against them prior to a fight and compare it to the rate at which they experience violent penalties after a fight within games. If fighting serves as a deterrent, we would expect to see fewer violent penalties within the game after the fight. Proponents of the deterrent argument would say that after a fight, the game calms down and fewer players take liberties with the opposing team out of greater fear that they would have to fight. Table 4 shows the empirical test of this claim.

For all games in which there was a fight, I conduct a t-test of difference in means between violent minors per minute prior to the fight and violent minors per minute after the fight. For games with multiple fights, I use the first fight to mark time before and after the fight. Sensitivity results that only examine games with one fight (86% of all games with fights) yield nearly identical results. Over the sample period, there were 2,842 games with a fight. The average number of violent minors per minute prior to the fight was 0.035, while it was 0.058 on average after a fight. This resulted in a difference of 0.023 fights more *after* a fight than before, which is

**Table 4. Difference in violent minor rate before and after fight.**

|  | Mean | Standard Deviation | T-statistic | p-value |
|---|---|---|---|---|
| Violent minor rate before fight | 0.035 | 0.078 |  |  |
| Violent minor rate after fight | 0.058 | 0.127 |  |  |
| **Difference** | **0.023** | **0.0028** | **8.35** | **0.0000** |

statistically significant at the 0.001% level. This means that instead of reducing the amount of violence in a game, fighting is associated with more violent penalties. Sensitivity analysis that drop fights in the last minute of the game that likely have little impact on the post-fight penalty rate yield nearly identical results. These results cannot rule out the claim that fights actually do have a deterrent effect on the team that loses the fight, but the implication would then be an even larger emboldening effect on the team that wins the fight, producing a net increase in violent play after the fight. This is certainly a possibility, but does not support the claim that fighting deters more violence on net, which is the question examined in this study.

Perhaps though, there is some deterrent effect across games rather than within games. That is, fighting may not deter violent behavior in the game in which the fight occurs, but only the next time that the teams face each other. To test this, I regress an indicator of whether a fight occurred between two teams in their previous game within a single season on the number of violent minors in the subsequent game. The models control for season-specific fixed effects for each two-team dyad. Each team is part of 29 different dyads (30 during 2017–19 when the league expanded to 31 teams), one with each other team in the league. For instance, the Anaheim Ducks played the Calgary Flames four times during the 2010–11 NHL season. The Ducks-Flames dyad is one of 435 unique two-team dyads for that season. These dyad fixed-effects control for unique circumstances that exist between the two teams that may influence the number of violent penalties taken in games between them, like whether the two teams are in the same division or faced each other in the playoffs the season prior. Models (2) and (3) add a seasonal time trend to control for differences in violent play or officiating as the season progresses. Model (3) includes additional controls for the number of days since the last time the two teams faced each other.

All specifications of the models in Table 5 show a negative coefficient estimate on the indicator of a fight in the previous game between the two teams, implying a fight in the previous game led to roughly 0.1 fewer violent minor penalties the next time the two teams meet. However, none of the coefficient estimates are statistically significant, suggesting it's unlikely fights have a meaningful deterrent effect across games. Even if there is a small between-game deterrent effect, the magnitude of the estimate suggests fighting is a highly inefficient way to deter violence. The coefficient estimate of -0.1 indicates that for every 10 fights there is one fewer violent minor penalty. Furthermore, because a full 35% of fights in the sample occurred in a game where the teams did not play each other again that season, there is no possibility those fights had a cross-game deterrent effect. Taken together, if fights in fact do deter violent behaviors *across* games at all—which is doubtful given the results in Table 5—on average it took roughly 15 fights to deter one violent minor penalty in the league. Interestingly, the number of days between games is negative and highly significant, suggesting more violent penalties in games where the two teams have recently played each other, though the magnitude is quite small.

### 3.4 The impact of individuals

I further investigate the question of whether teams can deter violent behaviors against them via fighting by considering whether certain individuals can prevent violence against their

**Table 5. Models of deterrence between games.**

|  | (1) | (2) | (3) |
|---|---|---|---|
| Fight in previous game | -.12 | -0.12 | -0.10 |
|  | (-1.13) | (-1.12) | (-1.00) |
| Days between games |  |  | -0.005*** |
|  |  |  | -3.31 |
| Dyad fixed effects | YES | YES | YES |
| Month of season time trend | NO | YES | YES |
| N | 6,858 | 6,858 | 6,858 |
| R-Squared | 0.52 | 0.52 | 0.52 |

t-statistic in parentheses;

*p<0.05,

**p<0.01,

***p<0.001

Clustered standard errors estimated

teams. Perhaps it's not the total number of fights a team has, but rather having a top fighter who could take on any opposing player in a fight that reduces the amount of violence against their team. The modern NHL presents an opportunity to evaluate this claim. The era of the one-dimensional enforcer-type player whose only purpose is to fight has passed. During the 1980's, 1990's, and early 2000's NHL teams carried certain players whose only skill was to fight. Faster game play, a greater emphasis on skill, and an increased supply of skilled players has eliminated this type of player from the NHL. Since fighting has become much scarcer and its impact on the game lessened, even players who are willing to fight have to offer something more than truculence to their teams. Consider that in the 2018–19 season, Tom Wilson and Michael Ferland were tied for the league lead in fights with six fights apiece, but each recorded 40 points and 22 and 17 goals, respectively. By comparison, for many seasons in the 1980's and 1990's the league's fighting leader had fewer than five goals and 15 points, but more than twenty fights.

This situation provides an opportunity to evaluate several natural experiments. In years past, an enforcer-type player would be inserted and removed from the lineup based on who their team is playing and whether the team was likely to need someone who can fight that game. In the modern NHL, players like Wilson, Ferland, and others serve the fighter role, but play every game because of their skill. They only leave the lineup if injured or suspended, providing exogenous variation where we can observe the same two teams playing with and without the potential deterrent (fighter) in the lineup and observe whether the opposing team takes more violent minors, majors, and misconducts against a team when their primary fighter is not in the lineup. This is particularly salient in the case of Michael Ferland, who was responsible for 6 of his team's 8 fights. Ferland's team—the Carolina Hurricanes—had no other credible threat to serve as a deterrent in their lineup when Ferland was out. I consider four such players for this part of the analysis: Tom Wilson, Michael Ferland, Austin Watson, and Patrick Maroon. Wilson, Ferland, and Maroon were tied for the most fights in the league in 2018–19 at six apiece and Watson had one fewer [5]. Each of these players has an established reputation as among the toughest fighters in the league and if it is possible for any single player to have a deterrent effect we would expect to see it among this group. In addition to each of these players being amongst the most frequent fighters in the league, each of them was not in the lineup for

**Table 6. Player specific effects on violent minors against and taken.**

|  | Wilson | | Ferland | | Maroon | | Watson | |
|---|---|---|---|---|---|---|---|---|
|  | Violent minors against | Violent minors taken | Violent minors against | Violent minors taken | Violent minors against | Violent minors taken | Violent minors against | Violent minors taken |
| Player in | -0.26 | 0.22 | 0.026 | -0.38 | 0.12 | 0.25 | -0.19 | -0.067 |
| lineup | (-0.92) | (0.66) | (0.10) | (-1.43) | (0.57) | (0.93) | (-0.70) | (-0.29) |
| Opposing team fixed effects | YES | YES | YES | YES | YES | YES | YES | YES |
| N | 82 | 82 | 82 | 82 | 82 | 82 | 82 | 82 |
| R-Squared | 0.55 | 0.28 | 0.37 | 0.44 | 0.37 | 0.32 | 0.45 | 0.37 |

t-statistic in parentheses; $^*p<0.05$, $^{**}p<0.01$, $^{***}p<0.001$

Clustered standard errors estimated

their respective teams for a substantial number of games due to injury or suspension. In the 2018–19 season, Wilson played in 63/82 games; Ferland played in 71/82 games; Maroon played in 74/82 games; Watson played in 37/82 games.

Table 6 shows the results of the linear fixed-effects regression of violent minors per game against a team on whether the players mentioned above were in the lineup for their team outlined in Eq 4. For these models, I use data from the 2018–19 season and the observations are team games. Each team played 82 games in that season. Opposing team fixed effects are included to control for any characteristics unique to each opposing team that might be correlated with violent minors against the team of interest.

In the case of each player, whether or not they were in the lineup did not have any statistically significant effect on the number of violent minors taken against their team. This provides plausibly causal evidence that having a player well-known for their fighting ability in the lineup does not reduce the number of violent minor penalties against his team. Additionally, having each of these players in the lineup does not cause a statistically significant increase in the number of violent minors their teams take. One weakness of this approach is the relatively low number of observations. To test the sensitivity of this result I perform the same regression but add data from the 2017–18 season to double the number of observations. This also introduces additional variation because both Michael Ferland and Patrick Maroon changed teams between the 2017–18 and 2018–19 seasons. Adding the additional season had no effect on the overall statistical significance or direction of the coefficients.

## 4. Discussion

On the surface, increasing the opportunities for traumatic brain injury by sanctioning bare-knuckle fighting as a punishment for other forms of potentially violent or dirty play seems contradictory, especially given recent advancements in our understanding about the short- and long-term impacts of traumatic brain injuries. However, if fighting serves to deter more violent behaviors, sanctioning it is justifiable given the opportunity cost of restricting NHL players' ability to police other player's actions. Both sides of the argument typically appeal to powerful anecdote to make their case. This study, systematically investigates the claim that fighting served as a deterrent to greater violence in the modern NHL era between 2010–19 by evaluating this hypothesis through a battery of different empirical tests. Not a single empirical approach yielded any evidence that fighting or even the threat of fighting deters violence in the NHL. To the contrary, the results suggest that fighting is strongly correlated with *more* violence—both at the team and individual game levels.

These results are consistent with Depken, Groothuis, and Strazicich [10], who find that changes in fighting behavior in the NHL over the past half-century were largely the result of changing norms within the sport, rather than league rule changes. Their study examines data from 1957 through 2013, roughly the beginning of the sample period in this study. Though the league did implement a rule requiring new players entering the league to wear protective visors for the 2013–14 season—which likely disincentivized fighting—the decline in both fighting and violent penalties preceded the rule by several seasons (Figs 1 and 2), reflecting broader changes in league norms.

More importantly, the results call into question the primary argument the league has used to defend its continued sanctioning of fighting, namely that fighting prevents greater violence against its players. While it is possible this may have been true for past generations of NHL players, this study presents substantial empirical evidence that refutes its veracity in the modern NHL context.

Although, this study is among the first to empirically investigate whether fighting has deterrent effects in the NHL, it is likely league executives have begun to suspect that fighting has little impact on other forms of violence. The deterrent argument may have been a socially acceptable cover for other less socially acceptable reasons to perpetuate fighting in hockey. The NHL and other North American revenue-producing leagues, such as the American Hockey League, East Coast Hockey League, Ontario Hockey League, Western Hockey League, Quebec Major Junior Hockey League, and United States Hockey League, have other incentives to continue to support fighting in hockey. Previous work has shown that fights may increase fan attendance [17, 18], which drive revenues in many of these leagues. This creates a conflict of interest for league executives that may place revenue growth above player health and safety. Perhaps league executives do place attendance and revenue growth above player safety, but in order to continue to put forth a deterrent argument, they must produce empirical evidence in support of it.

## Supporting information

**S1 Appendix.**
(DOCX)

**S1 Data.**
(DTA)

**S2 Data.**
(DO)

## Author Contributions

**Conceptualization:** Michael R. Betz.

**Data curation:** Michael R. Betz.

**Formal analysis:** Michael R. Betz.

**Funding acquisition:** Michael R. Betz.

**Investigation:** Michael R. Betz.

**Methodology:** Michael R. Betz.

**Project administration:** Michael R. Betz.

**Resources:** Michael R. Betz.

**Software:** Michael R. Betz.

**Supervision:** Michael R. Betz.

**Validation:** Michael R. Betz.

**Visualization:** Michael R. Betz.

**Writing – original draft:** Michael R. Betz.

**Writing – review & editing:** Michael R. Betz.

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
