## [Decision Letter · Decision Letter 0]

18 Apr 2022

PONE-D-22-06821Tooth for a Tooth: Does Fighting Serve as a Deterrent to Greater Violence in the Modern NHLPLOS ONE

Dear Dr. Betz,

Thank you for submitting your manuscript to PLOS ONE. After careful consideration, we feel that it has merit but does not fully meet PLOS ONE’s publication criteria as it currently stands. Therefore, we invite you to submit a revised version of the manuscript that addresses the points raised during the review process.

It is an interesting and well done study. Reviewers proposed some improvements: There are several mentions of (potential) medical consequences, yet medical data are completely absent from this paper. Coould yyou please give some comments on this porposal ? If a reduction in accidents and accident consequences could also be shown with the reduction in penalties, the arguments would be even more convincing.

The applied calculation models to the data collections should be explained already in the method, so the result presentation to the different relationship models can be shown more clearly. Currently the result presentation is a bit confusing, where different explanations and interpretations about this belong either in the introduction/method or discussion (especially e.g. individuals).

We look forward to receiving your revised manuscript.

Kind regards,

Hans-Peter Simmen, M.D., Professor of Surgery

Academic Editor

PLOS ONE

Journal Requirements:

Reviewers' comments:

Reviewer's Responses to Questions

**Comments to the Author**

1. Is the manuscript technically sound, and do the data support the conclusions?

Reviewer #1: Yes

Reviewer #2: Yes

2. Has the statistical analysis been performed appropriately and rigorously? 

Reviewer #1: Yes

Reviewer #2: Yes

3. Have the authors made all data underlying the findings in their manuscript fully available?

Reviewer #1: Yes

Reviewer #2: Yes

4. Is the manuscript presented in an intelligible fashion and written in standard English?

Reviewer #1: Yes

Reviewer #2: Yes

5. Review Comments to the Author

Reviewer #1: This study is very important and must be published. Even if ice hockey is known to be an emotional sport with physical contact, fights and unsporting behavior must be punished accordingly (see also IIHF rules and European ice hockey leagues) and are a bad example for the younger generation. However, the main argument in favor of fighting (deterrent effect of other violent behavior) can thus be refuted very well and objectively.

I have the following comments for the present study:

The data are based only on the penalty practices of the referees, who in turn have some leeway. Although the procedures are widely standardized, this provides some bias and may offer further potential for violence, especially in the case of incorrect decisions. The distinction made between tactical and violent penalties is often fuzzy in reality, but pragmatic for the study.

There are several mentions of (potential) medical consequences, yet medical data are completely absent from this paper. If a reduction in accidents and accident consequences could also be shown with the reduction in penalties, the arguments would be even more convincing.

The applied calculation models to the data collections should be explained already in the method, so the result presentation to the different relationship models can be shown more clearly. Currently the result presentation is a bit confusing, where different explanations and interpretations about this belong either in the introduction/method or discussion (especially e.g. individuals).

Otherwise, compliments and many thanks!

Reviewer #2: This paper is well done. You are analyzing a very important issue in the prevention of traumatic brain injuries, which are still too many. Compliment!

We have introduced in the past new rules in different ice hockey organization worldwide, like the IIHF, CAHA, etc. in order to reduce the number and severity of ice hockey related injuries, like the “checking from behind Rules” (CAHA in 1985, IIHF in 1994), the “head checking Rues” (IIHF in 2002), etc. However, the number of traumatic brain injuries are still too many and the referees are still not punishing head injuries sufficiently. We have still too many players, which are Fighting may not only cause fractures of the hands and face, lacerations and eye injuries, but may also cause very traumatic brain injuries, which may have long bad sequela to the players. Therefore, fighting should be completely abolished from the ice hockey arena and it should be punished more stronger not only in Europe, like we are doing now, but also in the NHL. NHL is the major goal and dream of every ice hockey players worldwide, therefore the NHL has to be an example for everybody. It is also important that fighting should also be punished very strongly in the minor hockey leagues as well.

Therefore, you have done a great job for the prevention of further ice hockey related traumatic brain injuries providing that fighting does not serve as deterrent against more violent behaviors. Compliment.

This paper should encourage every ice hockey community and first of all the NHL, to punish fighting more strongly in order to protect the safety of the players and to eliminate every not-necessary contact to the head from the ice arena. Principally head contact is not part of an ice hockey game, therefore every effort to eliminate head contact has only the potential to ameliorate the safety of our players!!

6. PLOS authors have the option to publish the peer review history of their article (what does this mean?). If published, this will include your full peer review and any attached files.

Reviewer #1: **Yes: **Walter Kistler

Reviewer #2: **Yes: **Dr. med. Nicola Biasca

Facharzt FMH für Orthopädische Chirurgie und

Traumatologie des Bewegungsapparates

Facharzt FMH für Chirurgie, Schwerpunkt Allg. Chirurgie und Unfallchirurgie

Fähigkeitsausweis für Sportmedizin SGSM

Past IIHF Medical Consultant

Staff Medical Team HCAP

---

## [Author Response · Author response to Decision Letter 0]

26 May 2022

Reviewer #1

Thank you for reviewing my study and your helpful comments. I’ve addressed your two main comments below.

Reviewer #1 Comment #1

The data are based only on the penalty practices of the referees, who in turn have some leeway. Although the procedures are widely standardized, this provides some bias and may offer further potential for violence, especially in the case of incorrect decisions. The distinction made between tactical and violent penalties is often fuzzy in reality, but pragmatic for the study.

There are several mentions of (potential) medical consequences, yet medical data are completely absent from this paper. If a reduction in accidents and accident consequences could also be shown with the reduction in penalties, the arguments would be even more convincing.

This is a good observation and I agree that ultimately the best measure of player safety is frequency and degree of injuries. However, this is particularly problematic in a study of the NHL because of the league’s restrictive injury disclosure policy. Unlike other major North American professional sports leagues (e.g. the National Football League and National Basketball Association) currently and during the period of study, the NHL’s injury disclosure policy allows teams to describe injuries simply as “lower body injury” or “upper body injury”. This means both a concussion and broken finger are listed as “upper body injuries”. Player medical records are closely guarded by the league and its member teams—and as you can imagine, this is particularly true for head injuries—thus, it’s doubtful the NHL or member teams would release such data for academic research. 

In absence of player medical records, I’ve endeavored to create a plausible proxy for violent game actions. My understanding of your main concern is that poor (or more lenient) officials may result in both fewer violent penalties being called and more fights, as players take matters into their own hands. This would lead to a downward bias in our estimates (e.g. the actual parameters are more positive than my estimates). I will point out several facts in response to this concern. First, while there is discretion among referees, the level of standardization—as you note—is high. NHL officials are full-time employees of the league and are subjected to rigorous training and performance review. Second, and perhaps more importantly, the design of the team level regressions (now Equation 1) aggregate penalties over the entire season. Throughout the course of an 82-game NHL season, the league uses an assignment mechanism that contains a random element to appoint two of the 44 full-time officials to each game. This randomness ensures teams are not disproportionately and systematically subjected to a particular referee throughout the course of a season, further reducing the likelihood of aggregate referee bias in our results. Finally, if referee-related bias is actually operative it would only strengthen the conclusions from our results. Each of the three coefficient estimates in Table 1 are positive and if the proposed bias is indeed operative and unaccounted for, then our results are actually an underestimate of the positive relationship between fights and violent penalties. 

Reviewer #1 Comment #2

The applied calculation models to the data collections should be explained already in the method, so the result presentation to the different relationship models can be shown more clearly. Currently the result presentation is a bit confusing, where different explanations and interpretations about this belong either in the introduction/method or discussion (especially e.g. individuals).

Thank you for the comment. I’ve included a Methods subsection spanning pages 8-9 in the revised text that details the estimation approach. Hopefully this will better prepare readers for the analysis and results. 

Reviewer #2

Thank you for reviewing my study and your encouraging comments. I hope that these empirical results will inform the debate over fighting’s role in the NHL.

---

## [Editor Report · Decision Letter 1]

30 May 2022

Tooth for a Tooth: Does Fighting Serve as a Deterrent to Greater Violence in the Modern NHL

PONE-D-22-06821R1

Dear Dr. Betz,

We’re pleased to inform you that your manuscript has been judged scientifically suitable for publication and will be formally accepted for publication once it meets all outstanding technical requirements. 

Kind regards,

Hans-Peter Simmen, M.D., Professor of Surgery

Academic Editor

PLOS ONE
---

## [Editor Report · Acceptance letter]

6 Jun 2022

PONE-D-22-06821R1 

Tooth for a Tooth: Does Fighting Serve as a Deterrent to Greater Violence in the Modern NHL 

Dear Dr. Betz:

I'm pleased to inform you that your manuscript has been deemed suitable for publication in PLOS ONE. Congratulations! Your manuscript is now with our production department. 

Kind regards, 

on behalf of

Dr. Hans-Peter Simmen 

Academic Editor

PLOS ONE